# Acute Ischemic Stroke Treatment in Germany (2015–2023): Nationwide Trends in Thrombolysis and Thrombectomy by Age and Sex

**DOI:** 10.3390/brainsci15080832

**Published:** 2025-08-02

**Authors:** Sara Hirsch, Karel Kostev, Christian Tanislav, Ali Hammed

**Affiliations:** 1Department of Geriatrics and Neurology, Diakonie Hospital Jung Stilling Siegen, Germany Wichernstrasse, 40, 57074 Siegen, Germany; sara.hirsch@diakonie-sw.de (S.H.); christian.tanislav@diakonie-sw.de (C.T.); 2Epidemiology, IQVIA, 60549 Frankfurt am Main, Germany; kostev@staff.uni-marburg.de

**Keywords:** acute ischemic stroke, intravenous thrombolysis, mechanical thrombectomy, stroke treatment trends, administrative data, Germany

## Abstract

Background: The implementation of intravenous thrombolysis (IVT) and mechanical thrombectomy (MT) in acute ischemic stroke (AIS) has proven effective, offering significant benefits for patient outcomes. We therefore investigated trends in the implementation of IVT and MT in Germany between 2015 and 2023. Methods: We conducted a retrospective analysis using German Diagnosis-Related Group (DRG) statistics from 2015 to 2023. Treatment numbers were analyzed annually based on OPS codes. We examined the age and sex distribution of patients undergoing these treatments. Additionally, we analyzed all hospital cases coded with ICD-10 for acute ischemic stroke (AIS). Results: Between 2015 and 2023, the number of AIS cases in Germany slightly declined from 250,802 to 248,107 (−1.1%), with the largest annual decrease (−4.3%) occurring during the COVID-19 pandemic (2019–2020). Despite this, the use of IVT increased from 40,766 cases (16.25%) in 2015 to 48,378 (19.50%) in 2023. MT usage rose even more sharply, from 7840 cases (3.13%) to 22,445 (9.05%). Among MT recipients, the proportion of patients aged ≥80 years rose significantly, from 27.2% to 42.1%. In this age group, women consistently comprised the majority of MT patients—65.4% in 2015 and 65.5% in 2023. Conclusions: Despite a stable stroke incidence, the use of IVT—and particularly MT—continued to increase in Germany from 2015 to 2023, even during the COVID-19 pandemic. MT usage nearly tripled, especially among patients aged ≥80 years. These trends highlight a resilient stroke care system and underscore the need for future planning to meet the rising demand for endovascular treatment in an aging population.

## 1. Introduction

Over the past two decades, significant advancements in the treatment of acute ischemic stroke (AIS) have been confirmed, leading to improved patient outcomes [1]. In 1995, intravenous thrombolysis (IVT) was established as an effective therapy for AIS [1]. The implementation of specialized departments—so-called Stroke Units (SUs)—also demonstrated clear benefits for patient outcomes. A further major development since 2015 has been the introduction of mechanical thrombectomy (MT) [1], which has since become standard treatment for patients with AIS and large artery occlusion of a cerebral vessel. It is estimated that approximately one-third of patients receiving IVT are also eligible for MT [2]. Five clinical trials have demonstrated significant benefits of MT in patients with AIS [1]. The pivotal study, MR CLEAN (2015), published by Berkhemer et al., involved a multicenter design including 500 patients from the Netherlands [3]. Almost all participants received standard medical therapy—specifically IVT (89%)—and were randomized to either an intervention group receiving MT (within six hours of stroke onset) or a control group receiving no endovascular treatment. The recanalization rate for MT was 59%, which was twice as high as in the control group, resulting in a clear clinical benefit [3]. Both treatment options—IVT and MT—are effective and, depending on the clinical scenario, can be applied separately or in combination [4]. A recent meta-analysis by Hammed et al., 2024 [4] reported comparable functional outcomes, mortality rates, and long-term recovery metrics between bridging therapy (IVT followed by MT) and MT alone. However, bridging therapy was associated with significantly greater early neurological improvement (as measured by NIHSS scores) and higher recanalization rates, both before and after thrombectomy [4].

The implementation of new treatment strategies into routine clinical practice typically occurs with a time lag. While the evidence base for IVT has existed for over two decades [1], MT is a more recent development [5,6,7]. For this reason, the present study investigates trends in the implementation of mechanical thrombectomy and intravenous thrombolysis in Germany between 2015 and 2023, using a national database that includes mandatory documentation of procedures such as IVT and MT.

## 2. Methods

### 2.1. Database

For this purpose, we conducted a retrospective analysis using the German Diagnosis-Related Group (DRG) statistics from 2015 to 2023. Treatment numbers were analyzed annually based on the Operation and Procedure Codes (OPS) for intravenous thrombolysis (8-020.8) and mechanical thrombectomy (8-836.80). This coding system is publicly accessible at www.icd-code.de (accessed on 8 February 2025). Additionally, we examined the age and sex distribution of patients undergoing these treatments. Age groups were categorized as follows: 0–19 years, 20–39 years, 40–59 years, 60–79 years, and 80 years and older. Sex distribution was divided into female and male patients.

Furthermore, we analyzed all hospital cases coded with ICD-10 I63 for acute ischemic stroke. Data for the years 2015 and 2016 were retrieved from the German Federal Statistical Office [8], while data from 2017 to 2022 were obtained via the DESTATIS website [9]. Based on these data, we calculated annual treatment rates for IVT and MT by comparing the number of patients receiving these interventions to the total number of ischemic stroke hospitalizations each year.

### 2.2. Study Population

The study sample included a total of 2,153,495 hospitalizations for acute ischemic stroke (ICD-10 code I63) in Germany between 2015 and 2023. Among these, 394,571 patients received intravenous thrombolysis (IVT) and 172,841 underwent mechanical thrombectomy (MT), as identified by OPS codes.

### 2.3. Statistical Analyses

This study was descriptive in nature and did not involve hypothesis testing. To assess trends over time, we calculated absolute frequencies and annual treatment rates for intravenous thrombolysis (IVT) and mechanical thrombectomy (MT) among patients with acute ischemic stroke (AIS) in Germany between 2015 and 2023.

The treatment rates were expressed as percentages of total documented AIS cases per year. Additionally, we determined relative changes in IVT and MT rates over time and stratified the data by age groups (0–19, 20–39, 40–59, 60–79, and ≥80 years) and sex (female, male). Special attention was given to trends in the oldest age group (≥80 years).

All statistical analyses were performed using IBM SPSS Statistics, Version 29.0 (IBM Corp., Armonk, NY, USA).

Time trends were visualized using line graphs for annual IVT and MT rates, as well as age- and sex-specific subgroup analyses.

## 3. Results

### 3.1. AIS Incidence over Time

In 2015, a total of 250,802 patients with acute ischemic stroke (AIS) were documented in Germany (Table 1). By 2023, this number had slightly declined to 248,107, representing a 1.1% overall decrease. The most pronounced reduction occurred during the COVID-19 pandemic, with a 4.3% drop observed between 2019 and 2020 (Figure 1). The female-to-male patient ratio remained stable over the entire period at approximately 0.9.

### 3.2. Intravenous Thrombolysis (IVT) Trends

The use of intravenous thrombolysis (IVT) showed a steady increase in both absolute numbers and relative proportions throughout the observation period. In 2015, 40,766 AIS patients (16.25%) received IVT, rising to 48,378 patients (19.50%) in 2023 (Figure 1). Stratification by age revealed a growing proportion of elderly patients: the proportion of IVT-treated individuals aged ≥80 years increased from 30.8% in 2015 to 35.3% in 2023 (Figure 2).

### 3.3. Mechanical Thrombectomy (MT) Trends

Mechanical thrombectomy (MT) demonstrated an even more pronounced upward trend. In 2015, 7840 AIS patients (3.13%) underwent MT, compared to 22,445 patients (9.05%) in 2023—nearly a threefold increase (Figure 1). This growth was consistent across all years of the observation period. The proportion of patients aged ≥80 years receiving MT increased from 27.2% in 2015 to 42.1% in 2023 (Figure 3).

### 3.4. Age- and Sex-Specific Trends in MT

Age-specific analysis showed that MT procedures among patients aged ≥80 years rose from 2131 in 2015 to 9468 in 2023—a 4.4-fold increase. In contrast, the number of MT cases in patients aged <80 years increased by a factor of 2.3 during the same period (from 5709 to 12,977) (Figure 4).

Throughout the study period, women consistently represented the majority of MT-treated patients in the ≥80 age group. In 2015, 65.4% (1394 of 2131) of MT patients in this age category were female, a proportion that remained nearly identical in 2023 (65.5%; 6203 of 9468) (Figure 4). This trend was stable across the entire observation period.

## 4. Discussion

### 4.1. Overview of Findings

This nationwide analysis of hospital data from 2015 to 2023 provides a comprehensive overview of trends in acute ischemic stroke (AIS) incidence and treatment in Germany. While the overall number of documented AIS cases declined only marginally over the nine-year period (−1.1%), there were sustained increases in the use of both intravenous thrombolysis (IVT) and mechanical thrombectomy (MT). IVT utilization rose from 16.3% in 2015 to 19.5% in 2023, while MT rates nearly tripled, increasing from 3.1% to 9.1%.

A notable demographic shift was observed among MT recipients: the proportion of patients aged ≥80 years increased from 27.2% in 2015 to 42.1% in 2023. Within this age group, women consistently comprised the majority of patients undergoing MT. These trends suggest an expansion of treatment eligibility and growing confidence in the safety and efficacy of MT in older populations.

Despite the challenges posed by the COVID-19 pandemic, stroke treatment rates continued to rise, reflecting the resilience and adaptability of the German stroke care system. These findings underscore the importance of ongoing investment in stroke care infrastructure, particularly in anticipation of a rising demand for endovascular therapies driven by an aging population.

### 4.2. Stroke Incidence During the COVID-19 Pandemic

The modest fluctuation in documented acute ischemic stroke (AIS) cases during the early phase of the COVID-19 pandemic aligns with international reports of reduced hospital presentations for acute vascular conditions [10,11,12,13,14,15]. Rather than indicating a true decline in stroke incidence, this variation likely stemmed from external factors such as patient reluctance to seek in-person care, fear of SARS-CoV-2 infection, and the temporary reallocation of healthcare resources toward pandemic-related services [16,17]. This trend was particularly evident in cases of transient ischemic attacks (TIAs), where brief and often less-alarming symptoms made these events more susceptible to underreporting or misclassification during periods of healthcare system strain [13].

Importantly, unlike many other disease areas that experienced significant care disruptions, the treatment of AIS in Germany remained largely unaffected. The utilization of both intravenous thrombolysis (IVT) and mechanical thrombectomy (MT) not only maintained pre-pandemic levels but continued to increase during the pandemic years. These findings underscore the resilience, adaptability, and preparedness of the German stroke care infrastructure, which ensured sustained access to time-critical interventions despite the unprecedented challenges posed by COVID-19.

### 4.3. Resilience of Acute Stroke Care Pathways

Despite a slight decline in overall acute ischemic stroke (AIS) cases, the delivery of time-critical treatments—intravenous thrombolysis (IVT) and mechanical thrombectomy (MT)—remained stable or even improved over the observation period. From 2015 to 2023, IVT rates increased from 16.3% to 19.5%, while MT rates nearly tripled, rising from 3.1% to 9.1%. Remarkably, these upward trends continued even during the COVID-19 pandemic, indicating that Germany’s stroke care infrastructure was sufficiently resilient and adaptable to sustain high-quality acute stroke treatment under unprecedented conditions.

This resilience likely reflects a well-coordinated system of care, characterized by the widespread availability of certified Stroke Units and consistent adherence to evidence-based protocols for AIS management [2,5]. Furthermore, the centralized organization of mechanical thrombectomy services may have contributed to stable treatment access, even as other non-urgent medical services were temporarily suspended.

### 4.4. Demographic Shifts in Treatment Patterns

Our age-stratified analysis revealed that the largest relative increase in MT utilization occurred among patients aged ≥80 years. The number of MT procedures in this age group rose from 2131 in 2015 to 9468 in 2023, representing a 4.4-fold increase. In contrast, MT rates in patients under 80 years of age increased by a factor of 2.3 over the same period (5709 to 12,977 cases). This shift indicates that clinicians are gaining increasing confidence in the use of MT in older patients over time. 

Sex-specific analysis showed that women represented the majority of MT-treated patients in the ≥80-year group (2015: 65.4%; 2023: 65.5%). This is consistent with the known epidemiology of stroke and vascular disease, where men tend to be affected at a younger age, while women experience strokes later in life [18,19]. Additionally, sex-specific coping strategies with illness could impact this result. Among patients under the age of 80, a higher proportion of MT procedures were performed in men (mean: 55.1%), though these differences did not affect the overall trend of increased MT utilization across both sexes and age groups. While access to treatment appeared equitable in terms of sex, these patterns warrant further investigation to identify and address potential disparities in stroke care delivery.

### 4.5. Expanded Indications and Evolving Clinical Practice

The increasing rates of mechanical thrombectomy (MT) in Germany can be attributed in part to evolving clinical practice and expanding treatment criteria. Landmark trials such as MR CLEAN [3] and meta-analyses like that of Hammed et al., 2024, [4] have firmly established the benefits of MT, including its efficacy in patients treated beyond conventional time windows or in combination with intravenous thrombolysis (IVT). Notably, the DAWN trial demonstrated the effectiveness of MT up to 24 h after symptom onset in selected patients, exhibiting a mismatch between clinical severity and infarct size [20]. These findings have significantly broadened eligibility criteria and supported the adoption of more aggressive reperfusion strategies, even among older patients and those with multiple comorbidities.

In parallel, the extension of the IVT treatment window from 3 to 4.5 h has increased the accessibility and applicability of thrombolytic therapy [1]. Furthermore, the combined use of IVT and MT—often referred to as bridging therapy—has demonstrated superior outcomes compared to MT alone, reinforcing the importance of integrated and coordinated stroke care models [4].

### 4.6. International Comparison and Potential for Further Growth

Although Germany has made substantial progress in the adoption of MT, there remains potential for further expansion. In 2023, MT was performed in approximately 9.1% of AIS cases—a notable achievement, but still lower than rates reported in countries like Slovakia, where MT reached 13% in 2020 [17]. This suggests that with continued investment in infrastructure, training, and regional accessibility, Germany could achieve even higher treatment penetration. Targeted efforts to reduce geographic disparities in access to MT—particularly in rural or underserved areas—could help optimize outcomes nationwide.

### 4.7. Implications for Health System Planning

The observed trends carry important implications for healthcare providers and policymakers. The consistent growth in mechanical thrombectomy (MT) utilization suggests that the procedure remains in an active phase of national rollout, with further potential for increased uptake. In contrast, intravenous thrombolysis (IVT) appears to be nearing a plateau, following more than two decades of integration into routine stroke care [1]. As the population continues to age and indications for MT expand, stroke networks must prepare for rising demand by ensuring adequate staffing, infrastructure, and effective interhospital transfer systems.

From an economic and systemic perspective, the COVID-19 pandemic may be viewed as a revealing stress test—or a “damasking” moment—that exposed potential overcapacities in parts of the healthcare system. While many outpatient and elective services were curtailed, time-critical procedures such as IVT and MT remained prioritized. Nonetheless, barriers to timely access during the pandemic likely contributed to increased morbidity and mortality in general [16]. The contribution of strokes to this observation remains unclear. Based on the results of our study, it can be assumed that the proportion of stroke patients who were unable to access thrombectomy and intravenous thrombolysis due to the pandemic have a minimal impact.

In this context, our findings show that IVT and MT usage did not follow the overall decline in stroke admissions during the pandemic. IVT treatment proportions increased slightly, from 16.3% in 2015 to 19.5% in 2023, with no substantial fluctuations during the pandemic years (mean 2019–2022: 18.6%, SD ± 0.02%). The growth in MT was more pronounced: absolute and relative numbers nearly tripled, from 7840 cases (3.1%) in 2015 to 22,445 cases (9.1%) in 2023. These consistent trends suggest that the German stroke care system maintained—and even expanded—access to time-sensitive interventions during a period of significant healthcare disruption.

This divergence also reflects the relative maturity of IVT, which has been well-established for over 25 years, compared to the more recent adoption of MT, which only became standard practice in 2015 [12]. As MT continues to be integrated into nationwide stroke pathways, further expansion is anticipated. This necessitates proactive planning to ensure equitable access to treatment and preparedness for future healthcare crises.

However, the dataset used in this analysis lacks important clinical parameters, including NIHSS scores, stroke onset times, imaging findings, and time-to-treatment intervals. As a result, it is not possible to assess stroke severity, treatment delays, or functional outcomes—all of which are essential for a comprehensive evaluation of quality of care.

### 4.8. Limitations

This study has several limitations. First, it relies on administrative data from the DRG system and OPS codes, which, despite routine auditing, are prone to potential coding inconsistencies, misclassification, and variation in documentation practices across institutions.

Second, the dataset lacks key clinical variables such as stroke severity (e.g., NIHSS scores), imaging results, and treatment timing metrics (e.g., onset-to-IVT/MT, door-to-needle times). As such, it is not possible to assess treatment appropriateness, workflow efficiency, or clinical outcomes, which are crucial for evaluating the quality of acute stroke care.

Third, no geographic information was available, preventing analysis of regional disparities in treatment access—this is particularly relevant for mechanical thrombectomy, where such differences have been reported in previous studies.

Fourth, the analysis is descriptive and observational in nature; it does not allow for causal inference or adjustment for confounding factors.

Finally, while our results show a growing proportion of elderly women receiving MT, we could not investigate whether this reflects purely demographic trends or underlying sex-specific differences in access, comorbidities, or clinical decision-making.

Despite these limitations, the study benefits from a large, comprehensive national dataset and provides meaningful insight into real-world trends in stroke treatment over nearly a decade.

### 4.9. Strengths

Despite these limitations, this study benefits from the use of comprehensive national data, covering nearly all hospital-based AIS cases in Germany over a nine-year period. The large sample size and consistent documentation provide robust insights into real-world treatment patterns. These findings can serve as a valuable reference for health system evaluation and future stroke care planning.

## 5. Conclusions

This nationwide study demonstrates that acute stroke care in Germany remained stable and resilient between 2015 and 2023, even during the COVID-19 pandemic. While the overall number of documented AIS cases showed minimal fluctuation, the use of intravenous thrombolysis (IVT) and especially mechanical thrombectomy (MT) increased significantly over the observation period. The rise in MT was particularly pronounced among patients aged ≥80 years, with women representing the majority in this group. These findings reflect not only the successful integration of MT into routine stroke care but also its growing application in elderly populations.

Importantly, the continued expansion of both IVT and MT during a global health crisis underscores the robustness of Germany’s stroke care infrastructure. As treatment indications broaden and demographic shifts increase demand, future health system planning must ensure adequate resources, personnel, and regional accessibility to maintain high-quality, equitable stroke care for all patients.

## Figures and Tables

**Figure 1 brainsci-15-00832-f001:**
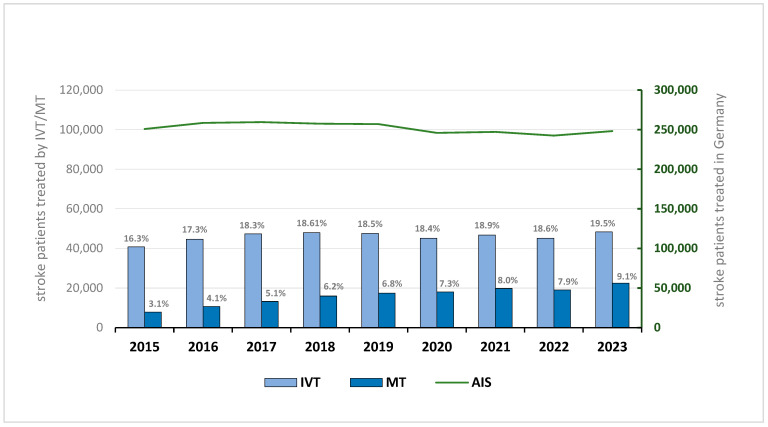
Trends in acute ischemic stroke (AIS) cases and treatment with intravenous thrombolysis (IVT) and mechanical thrombectomy (MT) in Germany from 2015 to 2023. The bar chart displays the number and percentage of AIS patients treated with IVT (light blue) and MT (dark blue) per year. The line graph (green) shows the total number of AIS cases recorded annually. While the overall AIS incidence remained relatively stable, IVT rates gradually increased and plateaued, whereas MT usage rose markedly, nearly tripling over the observation period. IVT = intravenous thrombolysis, MT = mechanical thrombectomy, AIS = acute ischemic stroke.

**Figure 2 brainsci-15-00832-f002:**
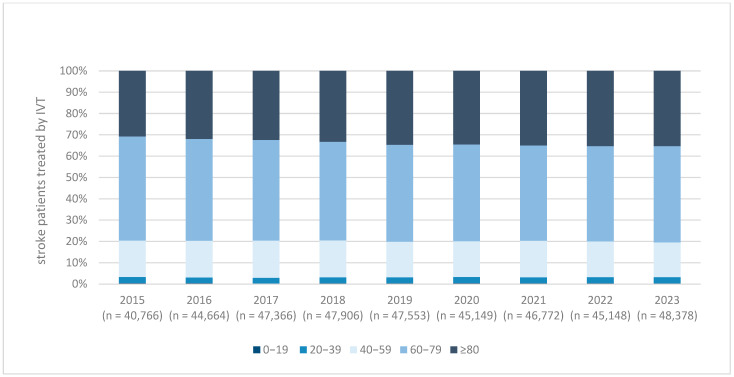
Age distribution in patients with ischemic stroke and treated by intravenous thrombolysis (IVT) from 2015 to 2023. Example: Age distribution of patients treated with intravenous thrombolysis (IVT) for acute ischemic stroke in Germany from 2015 to 2023. The stacked bar chart displays the relative proportions of stroke patients treated with IVT across five age groups (0–19, 20–39, 40–59, 60–79, and ≥80 years) for each year. The total number of IVT-treated patients per year is indicated below each bar. Over the observation period, the proportion of patients aged ≥80 years increased gradually, reflecting the growing application of IVT in older populations. IVT = intravenous thrombolysis.

**Figure 3 brainsci-15-00832-f003:**
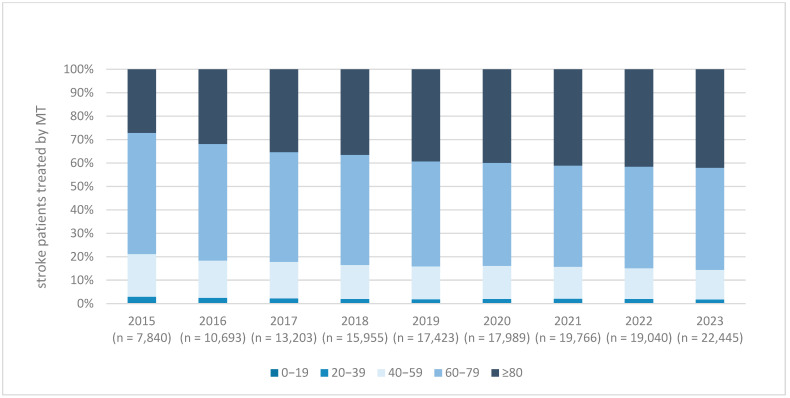
Age distribution in patients with ischemic stroke and treated by mechanical thrombectomy (MT) from 2015 to 2023. Age distribution of patients with acute ischemic stroke treated with mechanical thrombectomy (MT) in Germany from 2015 to 2023. The stacked bar chart displays the proportion of MT-treated patients stratified by age group: 0–19, 20–39, 40–59, 60–79, and ≥80 years. Each bar represents the percentage distribution within the total number of annual MT cases (indicated below each bar). Over time, the proportion of patients aged ≥80 years increased steadily, indicating broader use of MT in elderly populations. Abbreviation: MT = mechanical thrombectomy. MT = mechanical thrombectomy.

**Figure 4 brainsci-15-00832-f004:**
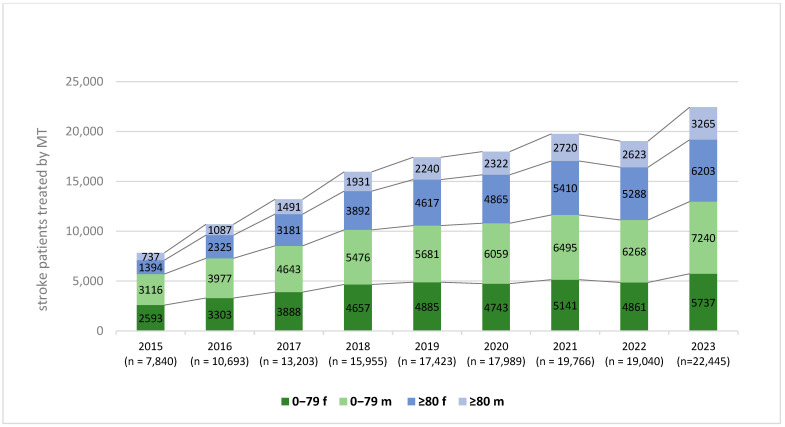
Age- and sex-specific trends in mechanical thrombectomy (MT) for acute ischemic stroke in Germany from 2015 to 2023. The stacked bar chart presents the absolute number of patients treated with MT each year, stratified by age group (<80 years and ≥80 years) and sex (female [f], male [m]). Over the observation period, a significant increase in MT procedures was observed, particularly among patients aged ≥80 years. Among this older age group, women consistently represented the majority of MT recipients. Total annual MT cases are indicated below each bar. MT = mechanical thrombectomy, f = female, m = male.

**Table 1 brainsci-15-00832-t001:** Annual absolute frequencies for acute ischemic strokes occurring in Germany between 2015 and 2023.

Year	2015	2016	2017	2018	2019	2020	2021	2022	2023
Annual absolute frequency	250,802	258,480	259,594	257,472	256,965	245,944	247,176	242,492	248,107

## Data Availability

The data presented in this study are available within the article. Additional details can be provided by the corresponding author upon reasonable request.

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
