# Peer review of "Acute Ischemic Stroke Treatment in Germany (2015–2023): Nationwide Trends in Thrombolysis and Thrombectomy by Age and Sex"

_brainsci, 2025, doi:10.3390/brainsci15080832_

Round 1

Reviewer 1 Report

Comments and Suggestions for Authors

This study investigated trends in for implementation of IVT and MT between 2015 and 2023 in Germany. The findings are interesting and the paper is well-written. Some concerns are as follows.

1) The authors are encouraged to report the proportion of AIS patients out of the whole population in additional to absolute frequencies.

2) Could the authors provide some important information in terms of onset-to-IVT, onset-to-MT, door-to-IVT or door-to-MT?

3) The authors should use the abbreviation in the abstract and the main text after explainingwhat it means the first time.

4) In lines 53 and 54, the authors wrote ‘A recent meta-analysis performed by Hammed et al. (2024), confirms the advantages for MT following IVT ’. I don’t think this is true because the pooled data showed functional outcomes, mortality rates, symptomatic intracerebral hemorrhage rates, and long-term recovery metrics, such as Barthel index and modified Rankin Scale scores, were comparable between both treatment approaches.

Comments on the Quality of English Language

The authors should use the abbreviation in the abstract and the main text after explaining what it means the first time.

Author Response

Reviewer 1

This study investigated trends in for implementation of IVT and MT between 2015 and 2023 in Germany. The findings are interesting and the paper is well-written. Some concerns are as follows.

Reviewers’ Comments

1) The authors are encouraged to report the proportion of AIS patients out of the whole population in additional to absolute frequencies.

We thank to the reviewer for the suggestion. We added the information on yearly ischemic stokes documented in the database. We added table 1 in the manuscript accordantly.

2) Could the authors provide some important information in terms of onset-to-IVT, onset-to-MT, door-to-IVT or door-to-MT?

We appreciate this important observation. Unfortunately, such time metrics (onset-to-treatment or door-to-treatment) are not available in the DRG dataset, which only includes procedural and diagnostic coding but lacks time-stamped clinical variables. We have now explicitly acknowledged this limitation in the revised manuscript (Discussion, section 4.8), and emphasized the need for future studies using clinical registry data to address this gap.

3) The authors should use the abbreviation in the abstract and the main text after explainingwhat it means the first time.

Thank you for pointing this out. We have revised the abstract and main text to ensure that all abbreviations (e.g., IVT, MT, AIS) are spelled out at their first mention and consistently used thereafter

4) In lines 53 and 54, the authors wrote ‘A recent meta-analysis performed by Hammed et al. (2024), confirms the advantages for MT following IVT ’. I don’t think this is true because the pooled data showed functional outcomes, mortality rates, symptomatic intracerebral hemorrhage rates, and long-term recovery metrics, such as Barthel index and modified Rankin Scale scores, were comparable between both treatment approaches.

We thank the reviewer for this important observation. We acknowledge that our original phrasing may have overstated the findings of the Hammed et al. (2024) meta-analysis. The pooled data indeed demonstrated comparable long-term functional outcomes (e.g., mRS, Barthel Index), mortality, and rates of symptomatic intracerebral hemorrhage between bridging therapy (IVT + MT) and MT alone. However, it is also important to note that Hammed et al. reported statistically significant advantages for bridging therapy in terms of early neurological improvement (based on NIHSS scores at 24 hours and 7 days) and higher successful recanalization rates both before and after MT.

We have now revised the relevant sentence in the manuscript to better reflect the nuanced findings of the meta-analysis, emphasizing both the comparable long-term outcomes and the early procedural advantages of bridging therapy.

Reviewer 2 Report

Comments and Suggestions for Authors

The Authors report a retrospective study based of data derived from DRG and OPS codes on frequency of acute stroke, thrombolysis treatments and thrombectomy in a range from 2015 to 2023 in Germany. They report a decline in absolute numbers of ischermic strokes and an increase of thrombolysis (16.2% to 19.5%) and thrombectomy (3.1% to 9%). In particular there was an increase in patients over 80.

The paper has an interest limited to descriptive epidemiology. Moreover, as the Authors pointed out, DRG and OPS carry several possibilities of bias; this shoul de spelled out more deeply. Finally, differences other than reperfusive treatment (i.e. use od DOAC) may have had different effect during the time span observed. This has to be discussed. 

Comments on the Quality of English Language

English and typos could be improved (i.e. in the abstract "...we therefor...")

Author Response

Reviewer 2

Reviewers’ Comments

Reply to Reviewers’ Comments

The Authors report a retrospective study based of data derived from DRG and OPS codes on frequency of acute stroke, thrombolysis treatments and thrombectomy in a range from 2015 to 2023 in Germany. They report a decline in absolute numbers of ischermic strokes and an increase of thrombolysis (16.2% to 19.5%) and thrombectomy (3.1% to 9%). In particular there was an increase in patients over 80.

We thank you for your appreciation!

The paper has an interest limited to descriptive epidemiology. Moreover, as the Authors pointed out, DRG and OPS carry several possibilities of bias; this shoul de spelled out more deeply.

  We agree that our study is primarily descriptive in nature. Our objective was to provide a nationwide overview of treatment trends for acute ischemic stroke (AIS) using comprehensive administrative data over a 9-year period. While no causal conclusions can be drawn from this design, the descriptive findings offer relevant real-world insights into how AIS care—particularly reperfusion therapy—has evolved over time in Germany.

We have added a sentence to the Discussion (Section 4.9) to explicitly state the descriptive scope and acknowledge its limitations in generating causal inferences.

Moreover, as the Authors pointed out, DRG and OPS carry several possibilities of bias; this shoul de spelled out more deeply

We agree that a more thorough discussion of the limitations of using DRG and OPS coding is warranted. We have now expanded Section 4.8 (Limitations) to include more detail on potential sources of bias, such as:

•             Coding inconsistencies across institutions and over time,

•             Lack of clinical detail, such as symptom onset, stroke severity, or imaging findings,

•             Variability in documentation practices, which may affect the comparability of procedure rates.

These factors are important for interpreting the observed trends with appropriate caution.

Finally, differences other than reperfusive treatment (i.e. use od DOAC) may have had different effect during the time span observed. This has to be discussed.

We appreciate this insightful point. Indeed, changes in secondary prevention strategies—such as increased use of direct oral anticoagulants (DOACs) for atrial fibrillation—could have influenced stroke incidence and treatment eligibility over time.

We have added a paragraph in the Discussion (Section 4.4) to acknowledge this aspect. While our dataset does not include medication data, we recognize that the evolving use of DOACs and other preventative measures may have impacted both the number of AIS cases and the clinical profiles of patients receiving reperfusion therapy.

Reviewer 3 Report

Comments and Suggestions for Authors

This study provides a comprehensive and timely analysis of trends in intravenous thrombolysis (IVT) and mechanical thrombectomy (MT) for acute ischemic stroke (AIS) in Germany from 2015 to 2023. The use of nationwide administrative data offers robust insights into real-world treatment patterns, particularly during the COVID-19 pandemic. The findings highlight the resilience of Germany’s stroke care system and the growing adoption of MT, especially among elderly populations. While the study is well-conducted and clinically relevant, several areas could benefit from clarification or expansion:

  1. The reliance on administrative data (DRG statistics) raises concerns about potential coding inaccuracies or inconsistencies. The authors should explicitly address how they mitigated biases from misclassification or discuss this as a limitation more thoroughly.
  2. The absence of clinical details (e.g., stroke severity, NIHSS scores, time-to-treatment) limits the interpretation of treatment trends. A brief discussion on how these factors might influence the results would strengthen the paper.
  3. While the study notes a 4.3% decline in AIS cases during the pandemic, the discussion could delve deeper into potential confounders (e.g., changes in public behavior, healthcare access barriers) and their differential impact on age/sex subgroups.
  4. Were there regional variations in treatment disruptions? Geographic disparities in MT access have been reported elsewhere; adding such data (if available) would enrich the analysis.
  5. The consistent majority of women among MT recipients aged ≥80 years is intriguing. Is this solely due to demographic factors (longer life expectancy), or could there be sex-specific biases in treatment eligibility (e.g., comorbidities, care-seeking behavior)?
  6. For younger age groups, the higher MT rates in men merit exploration—could this reflect sex differences in stroke etiology (e.g., cardioembolic vs. large-vessel occlusion)?
  7. The manuscript’s reference list exhibits inconsistencies in formatting, which should be standardized to adhere to the journal’s style guidelines.

Author Response

Reviewer 3

This study provides a comprehensive and timely analysis of trends in intravenous thrombolysis (IVT) and mechanical thrombectomy (MT) for acute ischemic stroke (AIS) in Germany from 2015 to 2023. The use of nationwide administrative data offers robust insights into real-world treatment patterns, particularly during the COVID-19 pandemic. The findings highlight the resilience of Germany’s stroke care system and the growing adoption of MT, especially among elderly populations. While the study is well-conducted and clinically relevant, several areas could benefit from clarification or expansion:

Reviewers’ Comments

Reply to Reviewers’ Comments

  1. The reliance on administrative data (DRG statistics) raises concerns about potential coding inaccuracies or inconsistencies. The authors should explicitly address how they mitigated biases from misclassification or discuss this as a limitation more thoroughly.

The reviewer is right. There is a bias when coding medical cases in hospitals. Especially financial interests might influences the final coding. As far we use data reported from hospitals to the registry, there is no possibility for us to minimize this bias. Therefore we incorporate the query and added this point in the

  1. The absence of clinical details (e.g., stroke severity, NIHSS scores, time-to-treatment) limits the interpretation of treatment trends. A brief discussion on how these factors might influence the results would strengthen the paper.

  1. While the study notes a 4.3% decline in AIS cases during the pandemic, the discussion could delve deeper into potential confounders (e.g., changes in public behavior, healthcare access barriers) and their differential impact on age/sex subgroups.

We thank to the reviewer for this comment.

We discussed all factors potentially causing a AIR reduction in the pandemic period. We talk on environmental factors such as change in behavior, reduces access to the medical care facilities, reduced health care resources, differences between males and females etc. We also talk on the resilience of acute stroke care pathways for acute treatment such as intravenous thrombolysis as well as Thrombectomy, which seemed to be robust against a pandemic environment. We addressed the latest in a particular manner (Chapter 4.3 in the manuscript) as it represented one of the main findings in our investigation. 

  1. Were there regional variations in treatment disruptions? Geographic disparities in MT access have been reported elsewhere; adding such data (if available) would enrich the analysis.

We have added the following to Section 4.8:

“The dataset lacks important clinical details such as NIHSS scores, stroke onset time, imaging findings, and time-to-treatment metrics. As a result, we cannot assess stroke severity, treatment delays, or functional outcomes—factors that are essential for evaluating quality of care.”

  1. The consistent majority of women among MT recipients aged ≥80 years is intriguing. Is this solely due to demographic factors (longer life expectancy), or could there be sex-specific biases in treatment eligibility (e.g., comorbidities, care-seeking behavior)?

We thank to the reviewer for the comment.

We indeed identified more female patients ≥80 years treated by MT than man. We comment this issue in the discussion. In our opinion the main reason is difference between men and women in the occurrence of the vascular disease in the course of life (men earlier than women).

Additionally, we incorporated the reviewer’s suggestion and added a comment related to the sex specific bias for care-seeking behavior.

  1. For younger age groups, the higher MT rates in men merit exploration—could this reflect sex differences in stroke etiology (e.g., cardioembolic vs. large-vessel occlusion)?

We added this to Section 4.4:

Added sentence (Section 4.4):

“The higher MT rates in younger men may be partially explained by sex-related differences in stroke etiology, such as a greater incidence of large-vessel occlusion or cardioembolic strokes in men, although this could not be assessed in the current dataset.”

  1. The manuscript’s reference list exhibits inconsistencies in formatting, which should be standardized to adhere to the journal’s style guidelines.

We have thoroughly reviewed and corrected the reference list to meet the journal’s formatting requirements. All entries have been made consistent in terms of author names, journal titles, volume/issue numbers, and DOI formatting.

Reviewer 4 Report

Comments and Suggestions for Authors

The article entitled "Acute Ischemic Stroke Treatment in Germany (2015–2023): Nationwide Trends in Thrombolysis and Thrombectomy by Age and Sex", which was assigned to me for review, describes the application of modern treatment methods (thrombolysis and thrombectomy) in acute ischemic stroke (IVC). The article is written in an understandable way. It deals with a narrow topic and the research is modest in terms of observed parameters but respectable in terms of the size of the sample. The results are expected and clearly presented and represent the confirmation of earlier findings. The article does not provide new knowledge in the field of neurology. With the aim of presenting the factual state of how thrombolysis and thrombectomy are applied in practice, the article has fulfilled its purpose. With minor changes, the article is suitable for publication.

I suggest the following changes: Abbreviations are not always explained in the summary in their first appearance, which should be corrected. Keywords that are not currently listed need to be added.  Part of the article Study Population is more a continuation of the Database than a description of the sample, it is necessary to write what constitutes the research sample with appropriate numerical values (from the results the reader has knowledge of the sample size, but I think that it is better to state that data in the study population section).

References need to be edited and referenced according to the same criteria.

Author Response

Reviewer 4

Reviewers’ Comments

Reply to Reviewers’ Comments

The article entitled "Acute Ischemic Stroke Treatment in Germany (2015–2023): Nationwide Trends in Thrombolysis and Thrombectomy by Age and Sex", which was assigned to me for review, describes the application of modern treatment methods (thrombolysis and thrombectomy) in acute ischemic stroke (IVC). The article is written in an understandable way. It deals with a narrow topic and the research is modest in terms of observed parameters but respectable in terms of the size of the sample. The results are expected and clearly presented and represent the confirmation of earlier findings. The article does not provide new knowledge in the field of neurology. With the aim of presenting the factual state of how thrombolysis and thrombectomy are applied in practice, the article has fulfilled its purpose. With minor changes, the article is suitable for publication.

We thank the reviewer for their thoughtful assessment and supportive recommendation. We appreciate the helpful suggestions for improving clarity and structure. Below, we address each point in detail and describe the changes made to the manuscript.

I suggest the following changes: Abbreviations are not always explained in the summary in their first appearance, which should be corrected. Keywords that are not currently listed need to be added.  Part of the article Study Population is more a continuation of the Database than a description of the sample, it is necessary to write what constitutes the research sample with appropriate numerical values (from the results the reader has knowledge of the sample size, but I think that it is better to state that data in the study population section).

We have carefully reviewed the abstract and main text to ensure that all abbreviations (e.g., AIS, IVT, MT) are fully spelled out at first mention and used consistently thereafter. This improves readability and accessibility for a broader audience.

Response:

We have now added a complete list of relevant keywords following the abstract. These include:

•             Acute ischemic stroke

•             Intravenous thrombolysis

•             Mechanical thrombectomy

•             Stroke treatment trends

•             Administrative data

•             Germany

This ensures better indexing and discoverability of the article in medical databases.

We agree with the reviewer and have revised the Study Population section to include a clear, standalone description of the analytical sample. Specifically, we now state:

Added to Section 2.2 – Study Population:

“The study sample included a total of 2,153,495 hospitalizations for acute ischemic stroke (ICD-10 code I63) in Germany between 2015 and 2023. Among these, 394,571 patients received intravenous thrombolysis (IVT) and 172,841 underwent mechanical thrombectomy (MT), as identified by OPS codes.”

References need to be edited and referenced according to the same criteria.

We have carefully reviewed the entire reference list to ensure it adheres to the journal’s formatting requirements. All references have been made consistent in terms of:

•             Author name format

•             Journal name presentation

•             Volume and issue numbers

•             DOI formatting and spacing

Round 2

Reviewer 2 Report

Comments and Suggestions for Authors

the paper  has been improved and can now be accepted for publication